# Suppressor effect of catechol-O-methyltransferase gene in prostate cancer

Yutaka Hashimoto[1,2], Marisa Shiina[1,2], Shigekatsu Maekawa[1,2], Taku Kato[1,2], Varahram Shahryari[1], Priyanka Kulkarni[1,2], Pritha Dasgupta[1,2], Soichiro Yamamura[1,2], Sharanjot Saini[1,2], Z. Laura Tabatabai[3], Rajvir Dahiya[1,2], Yuichiro Tanaka[1,2]*

1 Urology Section, Veterans Affairs Health Care System, San Francisco, CA, United States of America,
2 Department of Urology, University of California, San Francisco, CA, United States of America,
3 Department of Pathology, Veterans Affairs Health Care System and University of California, San Francisco, CA, United States of America

☯ These authors contributed equally to this work.
* Yuichiro.tanaka@ucsf.edu

**Data Availability Statement:** • TCGA PRAD miRNA and RNA sequencing data used in this study are available at the FireBrowse website (http://firebrowse.org). The direct links are below: - miRNA data: http://gdac.broadinstitute.org/runs/

## Abstract

Catechol-estrogens can cause genetic mutations and to counteract their oncogenicity, the catechol-O-methyltransferase (COMT) gene is capable of neutralizing these reactive compounds. In this study, we determined the functional effects and regulation of COMT in prostate cancer. Both the Cancer Genome Atlas (TCGA) and immunohistochemical analysis of clinical specimens demonstrated a reduction of COMT expression in prostate cancer. Also, western analyses of prostate cancer cell lines show COMT levels to be minimal in DuPro and DU145 and thus, these cells were used for further analyses. Re-expression of COMT led to suppressed migration ability (wound healing assay) and enhanced apoptosis (flow cytometric analyses), and when challenged with 4-hydroxyestradiol, a marked reduction of cell proliferation (MTT assay) was observed. Xenograft growth in athymic mice also resulted in inhibition due to COMT. As a mechanism, western analyses show cleaved CASP3 and BID were increased whereas XIAP and cIAP2 were reduced due to COMT. As COMT expression is low in prostate cancer, its regulation was determined. Databases identified several miRNAs capable of binding COMT and of these, miR-195 was observed to be increased in prostate cancer according to TCGA. Real-time PCR validated upregulation of miR-195 in clinical prostate cancer specimens as well as DuPro and DU145 and interestingly, luciferase reporter showed miR-195 capable of binding COMT and overexpressing miR-195 could reduce COMT in cells. These results demonstrate COMT to play a protective role by activating the apoptosis pathway and for miR-195 to regulate its expression. COMT may thus be a potential biomarker and gene of interest for therapeutic development for prostate cancer.

## Introduction

Prostate cancer is the most commonly diagnosed malignancy and the second leading cause of cancer death among aging men in the United States [1]. There will be an estimated 248,530

stddata__2016_01_28/data/PRAD/20160128/
gdac.broadinstitute.org_PRAD.Merge_mirnaseq__
illuminahiseq_mirnaseq__bcgsc_ca__Level_3__
miR_gene_expression__data.Level_3.
2016012800.0.0.tar.gz - mRNA data: http://gdac.
broadinstitute.org/runs/stddata__2016_01_28/
data/PRAD/20160128/gdac.broadinstitute.org_
PRAD.Merge_rnaseqv2__illuminahiseq_
rnaseqv2__unc_edu__Level_3__RSEM_genes_
normalized__data.Level_3.2016012800.0.0.tar.gz -
Clinical data: http://gdac.broadinstitute.org/runs/
stddata__2016_01_28/data/PRAD/20160128/
gdac.broadinstitute.org_PRAD.Merge_Clinical.
Level_1.2016012800.0.0.tar.gz Downloading data
from this site constitutes agreement to TCGA data
usage policy https://broadinstitute.atlassian.net/
wiki/spaces/GDAC/pages/844333156/Data+Usage
+Policy • COMT IHC data with normal prostate
epithelial tissues are available at the Protein Atlas
database (http://www.proteinatlas.org). The direct
links are below: -COMT IHC data with normal
prostate epithelial tissues https://www.proteinatlas.
org/ENSG00000093010-COMT/tissue/prostate#.

**Funding:** YT, I01BX000726, Department of
Veterans Affairs. RD, 1K6-BX004473, Department
of Veterans Affairs. The funders had no role in
study design, data collection and analysis, decision
to publish, or preparation of the manuscript.

**Competing interests:** The authors have declared
that no competing interests exist.

new cases and 34,130 deaths due to prostate cancer in the year 2021. This cancer is a disease of aging as 0.2% of persons will develop invasive prostate cancer before the age of 50, but drastically increases to 1.8% for those aged 50 to 59, 5.0% aged 60 to 69, and 8.7% aged 70 years and older. During their lifetime, 12.1% of men are likely to be diagnosed with prostate cancer [1].

The catechol-O-methyltransferase (COMT) gene is expressed in various cells of the body. COMT is involved in the inactivation of catecholamines such as dopamine, epinephrine, and norepinephrine by attaching a methyl group from the coenzyme S-adenosyl-L-methionine, to the hydroxyl group of the catechol, forming a methoxy compound [2]. However, aside from their function to degrade neurotransmitters, COMT also takes part in preventing the carcinogenesis process by inactivating reactive catechol compounds such as those formed during estrogen metabolism.

Estrogens are implicated in the prostatic carcinogenesis process. The normal prostate expresses aromatase that converts androgen to estrogen within the stroma but during malignancy, expression is increased in prostate cancer cells and thus, can lead to enhanced biosynthesis of estrogens in the local environment of the prostate [3]. Also, animal studies show evidence for a neoplastic role of estrogens in prostate carcinogenesis [4–7]. Estrogen on its own however, is considered weakly mutagenic or carcinogenic [8] and a more pronounced tumorigenic effect comes from its property to undergo oxidative pathways in cells to form highly reactive catechol metabolites such as hydroxyestrogen via cytochrome P450 (CYP) enzymes. These hydroxyestrogens undergo further oxidation to form semi-quinones and quinones and studies have shown these to react with DNA to form both stable adducts which remain in the DNA unless removed by repair; and larger amounts of depurinating adducts, which detach from DNA leaving behind apurinic sites [9–12]. Errors in the repair of these sites can then lead to critical mutations that initiate cancer. These types of mutations have been shown to occur for both estrone and estradiol [9–15]. Thus, agents produced in the estrogen metabolic pathway are demonstrated to be mutagenic. In prostate, formation of these estrogen metabolites and its role in cancer is supported by estradiol metabolites and DNA adducts found in urine [16,17].

Neutralization of metabolites such as hydroxyestrogen is thus critical to protect prostate cells from mutations and tumorigenesis. For this, COMT is the main cellular enzyme that quenches hydroxyestrogens and does so by conversion into inert methoxy compounds as mentioned above [18]. As a result, formation of highly reactive species and damaging catechol-quinones is prevented. These methoxyestrogens also have little or no affinity for estrogen receptors and have no estrogenic effects on target tissues [19].

The COMT gene is thus shown to play a suppressive role in cancer. However, in prostate, how COMT affects the carcinogenesis process is not understood as gene function studies are limited. In this report, we characterized the functional role and regulation of the COMT gene in prostate cancer cells. Our results are the first to show that the COMT gene plays a protective role in prostate cancer through the apoptosis pathway and that COMT is regulated by miRNA.

## Materials and methods

### Tissue specimens and sectioning

Two cohorts of samples of benign prostatic hyperplasia (BPH, n = 19 for immunohistochemistry (IHC) and n = 24 for miRNA PCR) and prostate cancer (n = 22 for IHC and n = 32 for miRNA PCR) were obtained from the Department of Anatomy and Pathology at the San Francisco Veterans Affairs Health Care System (SFVAHCS). Specimens are formalin-fixed, paraffin-embedded (FFPE) and de-identified data attached to patients who went through prostatectomy were obtained. Patients characteristics are shown in Tables 1 and 2 for IHC and

**Table 1. Patient characteristics of cohort of specimens at SFVAHCS for IHC analyses.**

| Variables | | BPH (n = 19) | Ca (n = 22) |
|---|---|---|---|
| Age (median ± SD) | | 69±7.06 | 64.5±7.39 |
| Pre-operative PSA (mg/l) (median ± SD) | | - | 5.1±4.68 |
| Gleason Score | | | |
| | <7 | - | 12 |
| | = 7 | - | 9 |
| | >7 | - | 1 |
| T stage | | | |
| | ≤2 | - | 17 |
| | >2 | - | 3 |
| | No record | - | 2 |

miRNA analyses, respectively. FFPE specimens were sliced into 5 μm sections and placed onto mounting slides using a microtome (Leica Biosystems, Buffalo Grove, IL) for further analyses as described below. This study was approved by the Clinical Research Office of the SFVAHCS and the Institutional Review Board of the University of California at San Francisco (UCSF).

## Cell lines and reagents

Human prostate cancer cell lines from brain metastatic DU145, bone metastatic PC3, lymph metastatic LNCaP, and non-malignant prostate epithelial cell line PWR-1E were purchased from American Type Culture Collection (Manassas, VA). DuPro cells derived from PC3 were also utilized [20] and obtained from collaborator Dr. R. Dahiya. Keratinocyte serum-free medium, bovine pituitary extract and human recombinant epidermal growth factor were purchased from Invitrogen (Carlsbad, CA). RPMI 1640, Opti-minimum essential medium and penicillin/streptomycin were obtained from Thermo Fisher Scientific, (Waltham, MA). Fetal bovine serum (FBS) was a product of Atlanta Biologicals (Lawrenceville, GA).

## Cell culture

DU145, DuPro,LNCaP, and PC3 cells were cultured in RPMI 1640 medium supplemented with 10% FBS and 1% penicillin/streptomycin. PWR-1E cells were cultured in keratinocyte growth medium supplemented with 5 ng/mL human recombinant epidermal growth factor,

**Table 2. Patient characteristics of cohort of specimens at SFVAHCS for miRNA PCR analyses.**

| Variables | | BPH (n = 24) | Ca (n = 32) |
|---|---|---|---|
| Age (median ± SD) | | 66.5±5.97 | 64.5±7.53 |
| Pre-operative PSA (mg/l) (median ± SD) | | - | 6.0±15.78 |
| Gleason Score | | | |
| | <7 | - | 16 |
| | = 7 | - | 11 |
| | >7 | - | 4 |
| | No record | | 1 |
| T stage | | | |
| | ≤2 | - | 17 |
| | >2 | - | 11 |
| | No record | | 4 |

0.05 mg/ml bovine pituitary extract and 1% penicillin/streptomycin. All cell lines were maintained at 37°C in a humidified atmosphere composed of 5% $CO_2$ and 95% air.

## Immunohistochemical analysis

Immunostaining of COMT was performed on specimens of BPH and prostate cancer (patient characteristics in Table 1). Slides consisting of tissue sections underwent the protocol of the UltraVision Detection System (Thermo Fisher Scientific, Waltham, MA) according to manufacturer's instructions. After 12-hour incubation with rabbit monoclonal antibody for COMT (1:500 dilution, #ab185954, Abcam, Cambridge, MA), 3, 3′-diaminobenzidine (DAB) was added as chromogen followed by counterstaining with hematoxylin. Cellular expression levels were analyzed by the intensity of positive cells using Fiji ImageJ [21]. The IHC images of normal prostate tissues were obtained from the Protein Atlas database (http://www.proteinatlas.org) [22].

## RNA extraction and quantitative real-time polymerase chain reaction

Tissue sections on slides (patient characteristics in Table 2) were scraped and from these and cell lines, total RNA was extracted using the miRNeasy kits (Qiagen, Valencia, CA) according to the manufacturer's instructions. RNA was then reverse-transcribed into complementary DNA (cDNA) using iScript™ RT Supermix for RT-qPCR (Bio-Rad, Hercules, CA). Levels of RNA expression were determined by quantitative real-time PCR analysis with the Applied Biosystems QuantStudio 7 System using TaqMan Universal PCR master mix according to the manufacturer's protocol (Applied Biosystems, Foster City, CA). PCR parameters for cycling were as follows: 95°C for 20 seconds, then 40 cycles of 95°C for 3 seconds and 60°C for 30 seconds. All reactions were done in a 10 μL reaction volume in triplicate. The data were analyzed using the delta-delta Ct method to calculate the fold-change. TaqMan probes (ThermoFisher Scientific, CA) and primers for COMT (assay ID: Hs00984972_g1), miR-15a (assay ID:000389), miR-16 (assay ID:000391), miR-195 (assay ID:477957), U48 (assay ID:001006), and GAPDH (assay ID: Hs02758991_g1) were obtained from Life technologies and Thermo Scientific. U48 and GAPDH were used as internal control.

## Western blot analysis

Whole cell extracts from cultured cells were prepared using RIPA Lysis and Extraction Buffer containing Halt™ Protease and Phosphatase Inhibitor Cocktail (Thermo Fisher Scientific, Rockford, IL). Protein quantification was done using a BCA protein assay kit (Thermo Fisher Scientific) according to the manufacturer's instructions. Samples were prepared with NuPAGE LDS sample buffer and NuPAGE® Sample Reducing Agent. Total protein (20–35 μg) was loaded onto NuPAGE® Novex 4–12% Bis-Tris Plus Gels with MES SDS Running Buffer with NuPAGE® Antioxidant and separated by XCell SureLock™ Mini-Cell Electrophoresis System (Thermo Fisher Scientific). Protein was transferred to iBlot® 2 Dry Blotting System and immunoblotting were carried out according to standard protocols. Monoclonal antibody against COMT was purchased from Abcam (#ab185954), whereas antibody for CASP3, cleaved CASP3, cleaved BID, XIAP, and cIAP2 were purchased from Cell Signaling Technology (Danvers, MA). Monoclonal antibody against β-Actin (Cell Signaling Technology) and GAPDH (Santa Cruz) were used to confirm equal loading. Protein complexes were visualized with Image Studio™ Software (LI-COR, Lincoln, NE) using the Odyssey® CLx Imaging System (LI-COR). The raw blotting images are available in S1 Fig.

## Stable and transient transfections

Stable COMT-expressing DuPro cells were created. Cells were transfected with pCMV6-EN-TRY vector expressing the C-terminally Myc and Flag-tagged human COMT cDNA or empty pCMV6-ENTRY vector as a control (OriGene Techologies, Rockville, MD) using FuGENE® HD Transfection Reagent (Promega, Madison, WI) according to the manufacturer's protocol. Clones were selected using 500 µg/ml of Geneticin® Selective Antibiotic (G418 Sulfate) (Thermo Fisher Scientific). Colonies resistant to G418 appeared within 2 weeks and colonies were picked and then expanded for another 3 weeks to make stable clone stock cells.

DU145 cells on the other hand, underwent transient transfections with COMT expressing plasmid (OriGene). Transfections were performed using FuGENE HD transfection reagent (Roche) according to the manufacturer's protocol. COMT knockdown was also performed. Three unique 27mer siRNA COMT duplexes (10 nM; OriGene Technologies, #SR319770) were transfected using lipofectamine 2000 transfection reagent in LNCaP cells and compared to control cells. Additionally, for overexpression of miR-195, mimic (25nM; Thermo Fisher Scientific, ID: AM10827) was transfected using JetPrime transfection reagent (Polyplus, Ill-kirch, France).

## Wound-healing assay

Migration ability of cells were measured by wound-healing assay. Cells were plated in 24-well plates containing inserts from CytoSelect Wound-Healing Assays (Cell Biolabs, San Diego CA) and monolayers allowed to form. The width of the initial gap (0 hour) and the residual gap 24 hours after wounding were calculated from photomicrographs taken using a Nikon Eclipse TS100 microscope (Technical Instruments, Burlingame CA). The wound areas and closures were measured using Photoshop Elements 15 (Adobe, San Jose CA).

## Apoptosis determination

Apoptosis was analyzed with an annexin V-fluorescein isothiocyanate (FITC)/propidium iodide (PI) staining system obtained from BD Biosciences (San Diego, CA). Briefly, prostate cells were harvested and resuspended in binding buffer at a concentration of $1 \times 10^6$ cells/ml. For each assay, $1 \times 10^5$ cells were incubated with annexin V-FITC and PI in the dark for 15 min at room temperature. After adding 400 µl of binding buffer, samples were analyzed using the BD FACSVerse flow cytometer (BD Biosciences, San Jose, CA).

## Estrogen metabolite treatment and cell proliferation assay

After plating COMT-expressing and control cells into 96-well plates at a density of $1 \times 10^3$ cells per well, growing cells were treated with 4-hydroxyestradiol (Millipore Sigma, St. Louis, MO) diluted in ethanol at concentration of 25 µM. After 24 hours, effects of metabolite on proliferation rates was determined using CellTiter Glo assay (Promega). At the desired time point, the number of viable cells were determined by adding CellTiter Glo Solution reagent to each well and measuring the luminescence by Victor X5 reader (PerkinElmer, Waltham MA). Results were expressed as relative cell number compared to DMSO control.

## *In vivo* tumor growth

All animal care was in accordance with the guidelines and this study was approved by the IACUC (Institutional Animal Care and Use Committee) for experimentation at the San Francisco Veterans Affairs animal facility which is accredited through AAALAS (American Association for Accreditation of Laboratory Animal Science). Animal users completed training

programs to handle and work with mice prior to animal experiments. A subcutaneous xeno-graft mouse model was utilized. DuPro cells ($5 \times 10^6$) stably transfected with COMT or empty pCMV6-ENTRY vector established above were suspended in 100 μL Matrigel (Corning, Corning, NY) and subcutaneously injected into the backside flank of five-week-old male nude mice (strain BALB/c nu/nu; Charles River Laboratories, Wilmington, MA). Four animals were used per treatment group and tumor growth was examined after 21 days. Tumor volume was calculated on the basis of width (x) and length (y) using the formula: $x^2y/2$, where $x < y$.

## Luciferase assay

Reporter vectors were constructed by ligation of annealed custom oligonucleotides containing the putative target binding sites of S-COMT 3'-UTR into pmiR-GLO reporter vector (Promega). The binding site was searched using TargetScan. 293T cells were transfected with pmiR-GLO vector containing the COMT 3'-UTR sequences (oligo seq) and miR-195 mimic (ID: AM10827) with lipofectamine 2000 reagent. Target site deletion was used as control. Luciferase activity was measured 48 hours after transfection using a Dual-Luciferase Reporter Assay System (Promega). Relative luciferase activity was calculated by normalizing to renilla luminescence.

## Statistical analysis

Values are presented as the mean ± standard deviation (SD) based on results obtained from at least three independent experiments. For *in vivo* studies, results are based on four animals per group. The relationship between variables was analyzed using the non-parametric Mann-Whitney U test, two-tailed Student's t-test, Dunnett's test, or one-way analysis of variance (ANOVA). Overall survival time endpoints were defined from the time of surgery until time of death or last follow-up. Receiver Operating Characteristic (ROC) curve was used to compute cut-off value for patient's survival time analysis and log rank test was used to calculate *p*-values in overall survival time analyses. All analyses were performed using GraphPad Prism (GraphPad Software, San Diego, CA) or R [23].

# Results

## COMT expression in prostate cancer patients and cell lines

Expression levels of COMT were initially evaluated in prostate cancer cases and non-cancerous controls by utilizing the TCGA database (The data is available at 'http://firebrowse.org'). Analysis of 51 controls and 499 cases show that levels are generally lower in tumors although significance was not achieved (*p* = 0.2, Fig 1A). However, when evaluated by progression, higher stages (>T3, n = 290) had significantly lower levels compared to T2 (n = 185, *p* = 0.007, Fig 1B) and higher Gleason score (≥GS4+3; n = 294) showed slightly lower expression than lower Gleason (≤GS3+4; n = 188) although not significant (*p* = 0.2, Fig 1C). Interestingly when determining survival probability, low expressing patients (n = 171) had significantly reduced survival rates compared to those with high expression (n = 311) (*p* = 0.0026, Fig 1D). Levels of COMT protein were also lower in prostate cancer compared to BPH as determined by immunohistochemical analysis of clinical specimens (*p* = 0.004, Fig 1E). Positive COMT expression was observed in luminal cells and basal region in normal and BPH samples (S2 and S3 Figs). Cell lines were then evaluated for expression. As shown in Fig 1F, levels of COMT RNA were reduced in AR negative DU145, DuPro and PC3 cells compared to normal PWR-1E cells (*p* < 0.001) although was higher in AR positive LNCaP cells. Western analyses also showed protein levels to be reduced or minimal in AR negative but not AR positive cells (Fig

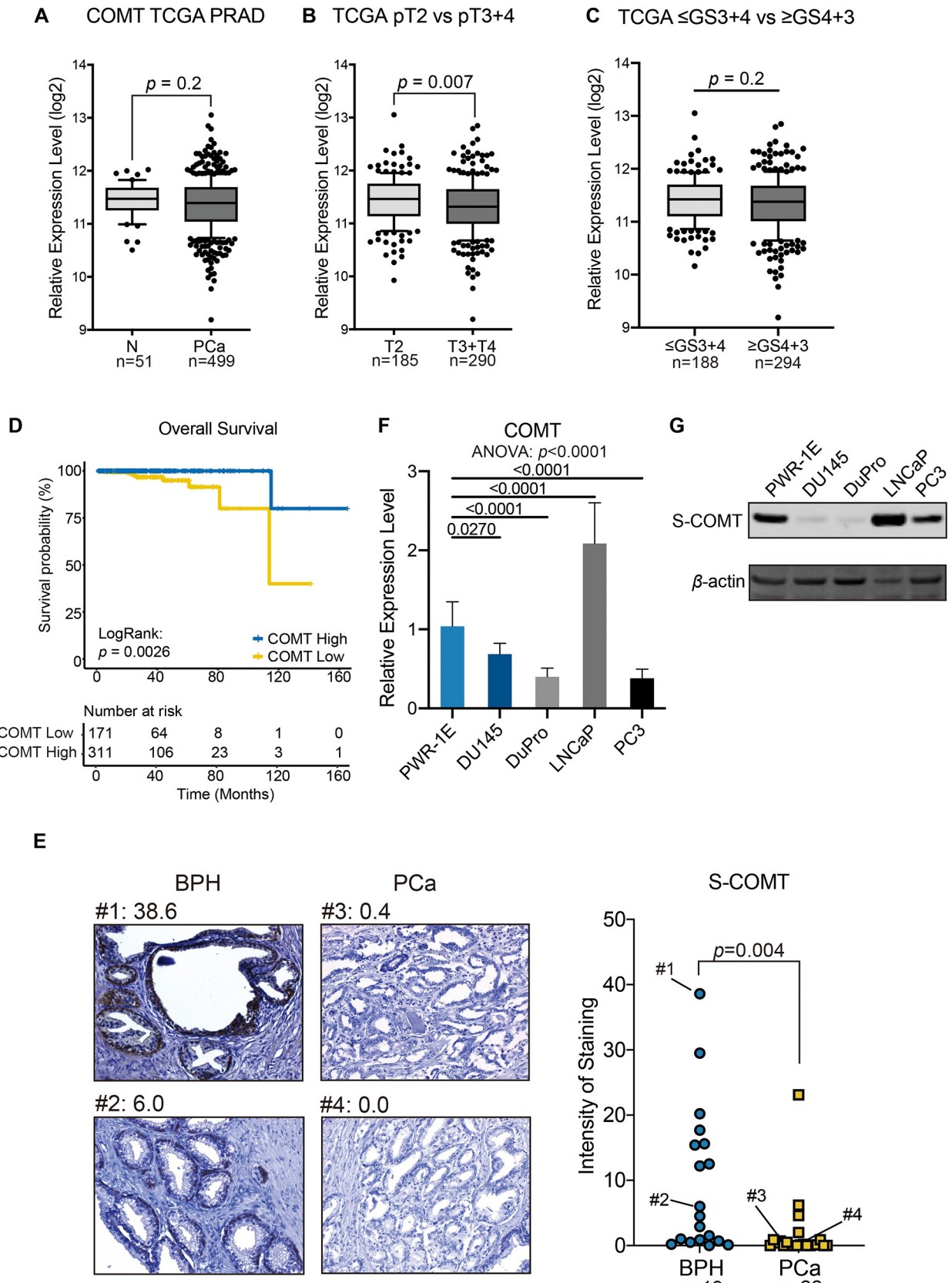

**E**

BPH

#1: 38.6

#2: 6.0

PCa

#3: 0.4

#4: 0.0

**Fig 1. COMT is downregulated in advanced prostate cancer tissues and cell lines.** Analysis of TCGA data show (A) COMT levels to be lower in cancerous (n = 499) versus normal (n = 51) tissues ($p$ = 0.2), (B) COMT levels to be lower in higher stages (T3+T4, n = 290 versus T2, n = 185) of cancer ($p$ = 0.007), and (C) COMT levels to be lower in higher Gleason score group ($\geq$GS4+3, n = 294 versus $\leq$GS3+4, n = 188) of cancer ($p$ = 0.2) (D) Kaplan-Meier curves show lower levels (n = 171) of COMT to lead to poorer overall survival compared to the group showing higher levels (n = 311) (Log rank test: $p$ = 0.0026). (E) COMT protein by immunohistochemical analysis in clinical specimens. *Left*: Example of clinical specimen staining. Numbers above images reflect intensity of staining. *Right*: Staining is lower in specimens of prostate cancer (n = 22) compared to BPH (n = 19). (F-G) COMT expression is downregulated in AR negative prostate cancer cell lines (DU145, DuPro, and PC3) compared to AR positive PWR-1E and LNCaP cells. Cells were grown in culture dishes for 48 hours. (F) Relative mRNA levels of COMT in cell lines as determined by real-time PCR. Expression levels are normalized to PWR-1E. The $p$-values are based on Dunnett's multiple comparison test (shown for each comparison) following one-way ANOVA test ($p$-value shown below graph title). (G) Representative immunoblot displaying COMT protein expression in cell lines. $\beta$-actin was used as loading control. The middle bar in the boxplot shows median and lower and higher whiskers shows 10th and 90th percentile, respectively. Data are presented as mean ± SD. The $p$-values are based on Mann-Whitney U-test in A-C and E.

1G). These results indicate that non-malignant tissues such as normal and BPH show higher COMT expression level with AR negative prostate cancer cells having reduced COMT levels.

## Effect of COMT on cell growth properties

We observed reduced levels of COMT in DuPro and DU145 cells and tested if COMT overexpression impacts the properties of these cells. DuPro stably expressing COMT were developed and DU145 transiently expressing COMT were utilized and protein levels of COMT in these cells were confirmed to be increased. On the other hand, COMT expression levels were observed to be low or absent in mock and vector controls (Fig 2A). Expression of COMT led to inhibition of migration ability of cells as determined by wound healing assay. Rate of closures compared to control due to COMT were 0.727 for DuPro ($p$ = 0.0003, Fig 2B) and 0.537 for DU145 after 24 hours ($p$<0.0001, Fig 2B). Also, COMT significantly induced apoptosis after 48 hours as total levels were 3.5% versus 18.0% in DuPro ($p$ = 0.0045) and 6.4% versus 16.4% in DU145 ($p$ = 0.0003) for pCMV versus COMT-expressing cells, respectively (Fig 2C). Additionally, since COMT is capable of reducing oncogenic catechol compounds, we determined COMT effects on cells in the presence of 4-hydroxyestradiol. As shown in Fig 2D, COMT dramatically reduced proliferation of cells by 94% in DuPro ($p$ = 0.0005) and 35% in DU145 ($p$ = 0.0114) after 24 hours. These results thus indicate COMT to play a role in impeding cell progression properties. In contrast, COMT knockdown using siRNAs confirmed that COMT downregulation increased migration capability and decreased cell apoptosis (S4 Fig).

## COMT effect on tumor formation *in vivo*

The effects of COMT overexpression on cells *in vitro* were corroborated using murine xenograft model by injecting stable COMT-expressing and vector control DuPro cells subcutaneously. In concordance, COMT expression abrogated xenograft tumor formation and this inhibition continued throughout the experimental period of three weeks. In contrast, tumors in pCMV mice were visible by week 1 and growth was markedly enhanced by week 3 as the average tumor volume grew to 457 mm3 in controls whereas in comparison, stable COMT cells grew to only 5.6 mm3 in mice ($p$<0.0049) (Fig 3). Thus, prostate cancer cell xenograft growth *in vivo* is repressed due to COMT.

## COMT affects apoptosis-related genes

Enhancement of apoptosis due to COMT in Fig 2C implies the involvement of pathway genes. CASP3 and BID are known to increase apoptosis whereas XIAP and cIAP2 are known to prevent apoptosis [23]. Expression levels of these genes were thus determined in COMT-expressing DuPro cells. Western analyses show cleaved CASP3 and BID to be upregulated with XIAP

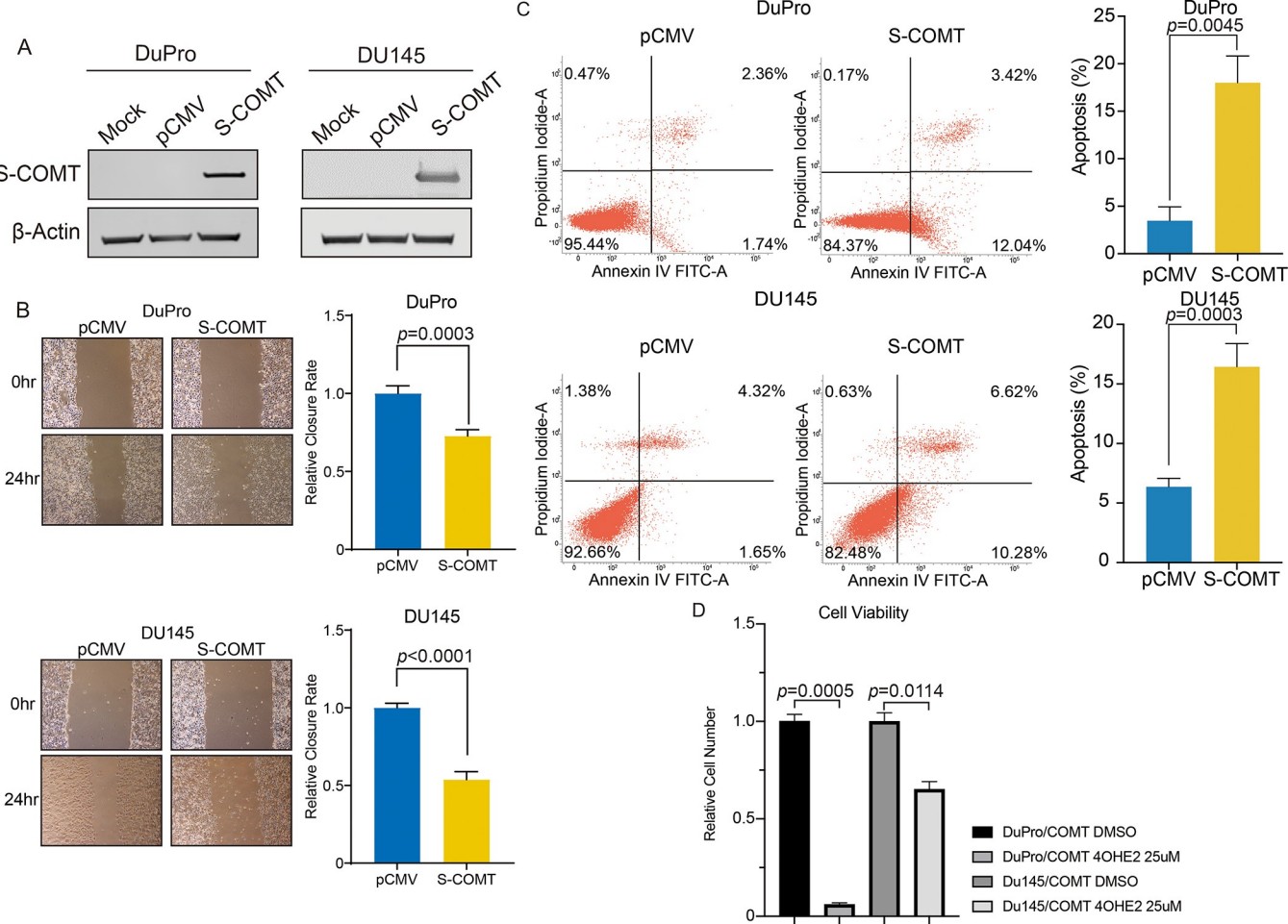

**Fig 2. Re-expression and tumor suppressive effect of COMT on cells.** (A) Ectopic expression of COMT. DuPro cells stably transfected and DU145 cells transiently transfected with either COMT or empty vector (pCMV) along with mock were grown for 48 hours and underwent western analyses. β-actin was used as loading control. (B) Cell migration as measured by wound healing assay. CytoSelect Wound Healing assay was used, and closure measured after 24 hours. *Left*: Representative images of wound healing assay are shown. *Right*: Migration expressed as relative closure rate of wound normalized to pCMV. (C) COMT expression upregulates apoptosis. Stable COMT-expressing DuPro and transiently COMT-expressing DU145 cells along with vector controls were grown for 72 hours and apoptosis was measured by flow cytometric analyses. *Left*: Representative quadrant dot plot showing cell population in early (bottom right quadrant) and late (top right quadrant) apoptotic states for each treatment. *Right*: Total apoptosis %. (D) COMT expression attenuates 4-hydroxyestradiol stimulated proliferation. Stable COMT-expressing DuPro and transient COMT expressing DU145 were treated with 4-hydroxyestradiol at a dosage of 25 μM for 24 hours, and cell proliferation was analyzed by the MTS proliferation assay. Results are normalized to DMSO control. Data are presented as mean ± SD. All *p*-values are based on paired t-test.

and cIAP2 being downregulated due to COMT compared to pCMV control (Fig 4). It is thus apparent that these genes mediate COMT effects on apoptosis in prostate cancer cells.

## COMT regulation by miR-195

Since COMT levels are reduced in prostate cancer as shown in Fig 1, we then evaluated its regulation through miRNA. Bioinformatics databases miRDB, miRmap, miRWalk and TargetScan were utilized and common to these were 9 miRNAs capable of binding COMT (Fig 5A). Of these, five miRNAs (miR-15a, miR-16, miR-195, miR-497, miR-939) were upregulated in cancer compared to normal prostate according to TCGA database (Fig 5B). We validated three of these miRNAs (miR-15a, miR-16, miR-195) in cells by real-time PCR and found miR-195 to be highly expressed in both parental DuPro ($p = 0.0008$) and DU145 ($p = 0.01$) compared to

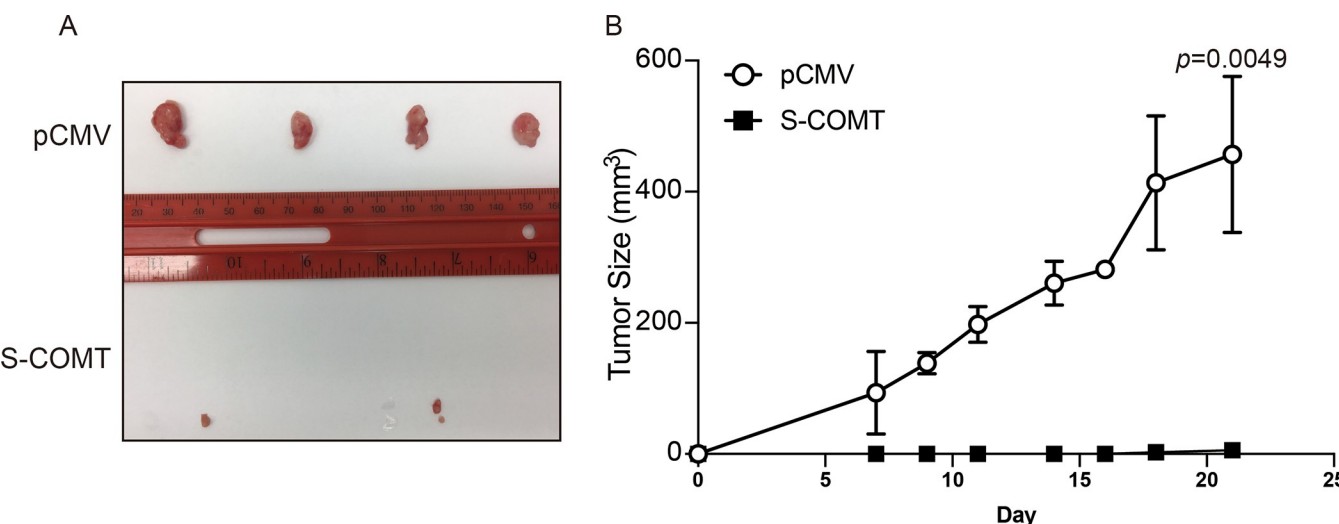

**Fig 3. Expression of S-COMT suppresses xenograft growth of cells *in vivo*.** Four athymic nude mice were injected subcutaneously with stable S-COMT (left flank) or pCMV-transfected (right flank) DuPro cells and growth determined over time. (A) Tumors extracted from mice after 21 days. (B) Tumor size (mm³) over time. Data are presented as mean ± SD. The *p*-value is based on paired t-test.

normal PWR-1E cells (Fig 5C). miR-497 and miR-939 were excluded since sample medians of these miRNAs showed low RPM (<50) in TCGA. Analysis on clinical specimens also show significantly higher miR-195 expression in prostate cancer compared to BPH tissues (*p*<0.0001, Fig 5D). Further, regression analysis between COMT and miR-195 in TCGA show a significant R value of -0.16 (*p* = 0.0005, Fig 5E).

We next carried out luciferase reporter assays to determine direct binding between miR-195 and COMT. Reporter vectors consisting of the miR-195 binding site and its deletion on the COMT 3'-UTR were created (Fig 6A). As shown in Fig 6B, in the presence of miR-195 mimic, the reporter vector harboring the miR-195 binding site had reduced activities compared to those with deletion vectors (p = 0.0036). We also overexpressed miR-195 using mimic in PWR-1E, LNCaP and PC3 cells and confirmed miR-195 could decrease COMT expression (Fig 6C). These results thus indicate miR-195 to interact and regulate COMT expression levels in prostate normal epithelial and cancer cells. Additionally, we found higher miR-195 expression levels could relate to poorer overall survival in prostate cancer patients according to

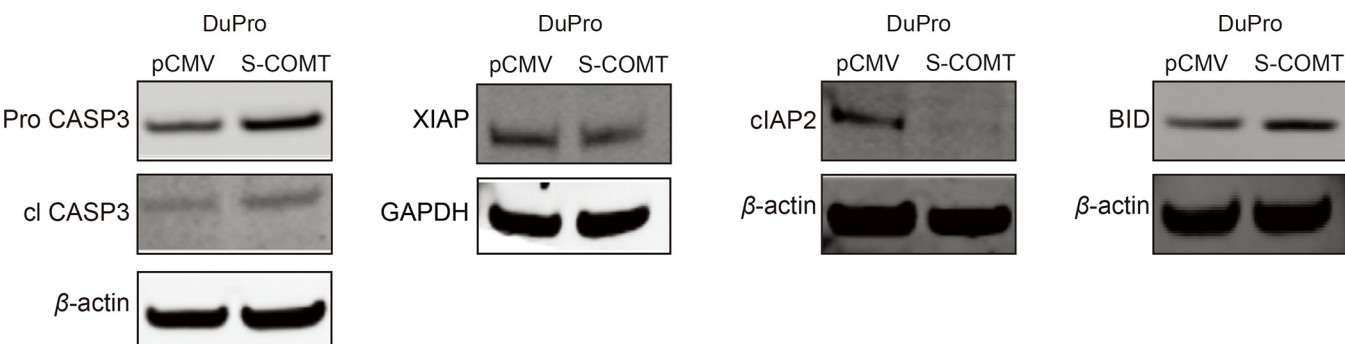

**Fig 4. COMT affects apoptosis related genes in cells.** Stable COMT-expressing and vector control DuPro cells were grown and expression of genes were determined by western analyses. Cleaved CASP3 and BID were observed to be increased whereas XIAP and cIAP2 genes were reduced. β-actin and GAPDH was used as loading control.

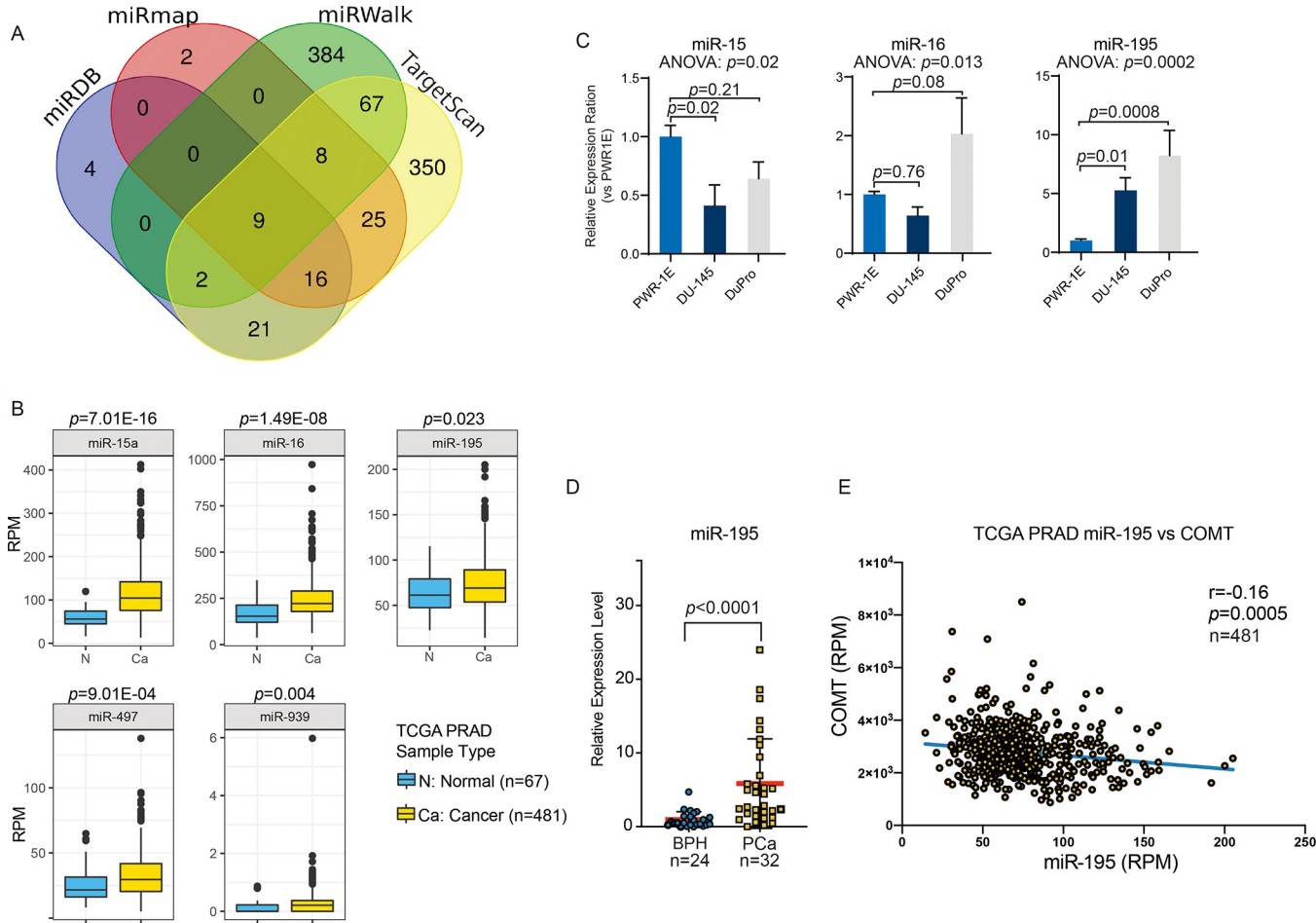

**Fig 5. Associations between COMT and miRNA.** (A) Venn diagram consisting of four databases (miRDB, miRmap, miRWalk, TargetScan) indicating 9 possible miRNAs capable of binding COMT gene. (B) Of candidate miRNAs identified by all four databases to bind COMT, five showed oncogenic effect according to TCGA. (C) Of these five, miR-195 is the only miRNA determined to be upregulated in both parental DuPro and DU145 compared to PWR-1E cells. MiR-497 and miR-939 were not analyzed due to low RPM (<50) in TCGA. The *p*-values are based on Dunnett's multiple comparison test (shown for each comparison) following one-way ANOVA test (*p*-value shown below graph title). (D) miR-195 was also upregulated in clinical specimens of prostate cancer (n = 32) compared to BPH (n = 24) as determined by real-time PCR. (E) Spearman nonparametric correlation analysis between COMT versus miR-195 expression in TCGA. The slope shows simple linear regression Data are presented as mean ± SD. The *p*-values are based on Mann-Whitney U-test in B and D.

TCGA (*p* = 0.013, Fig 6D) and is consistent with reduced COMT expression levels leading to lower overall survival rate in the same cohort (Fig 1D).

## Discussion

The bioactivation of estrogens into catechol-estrogens such as hydroxy-estrogens through CYPs have been shown to increase proliferation of cells [24,25] and associate with tumors including prostate [9,26]. These forms of estrogen can subsequently lead to the formation of semi-quinones and quinones that are known to react with DNA [27,28]. Damaging reactive oxygen species are also produced in this process and not only do they cause oxidative DNA damage but also promote neoplastic transformation of initiated cells [29]. Further, hydroxyestradiol has been observed to induce CYP1B1-mediated oncogenicity [30], suggesting a feedback loop. Therefore, elimination of catechol-estrogens in cells of the body is important to prevent the carcinogenesis process and the COMT enzyme is responsible for this defense. We

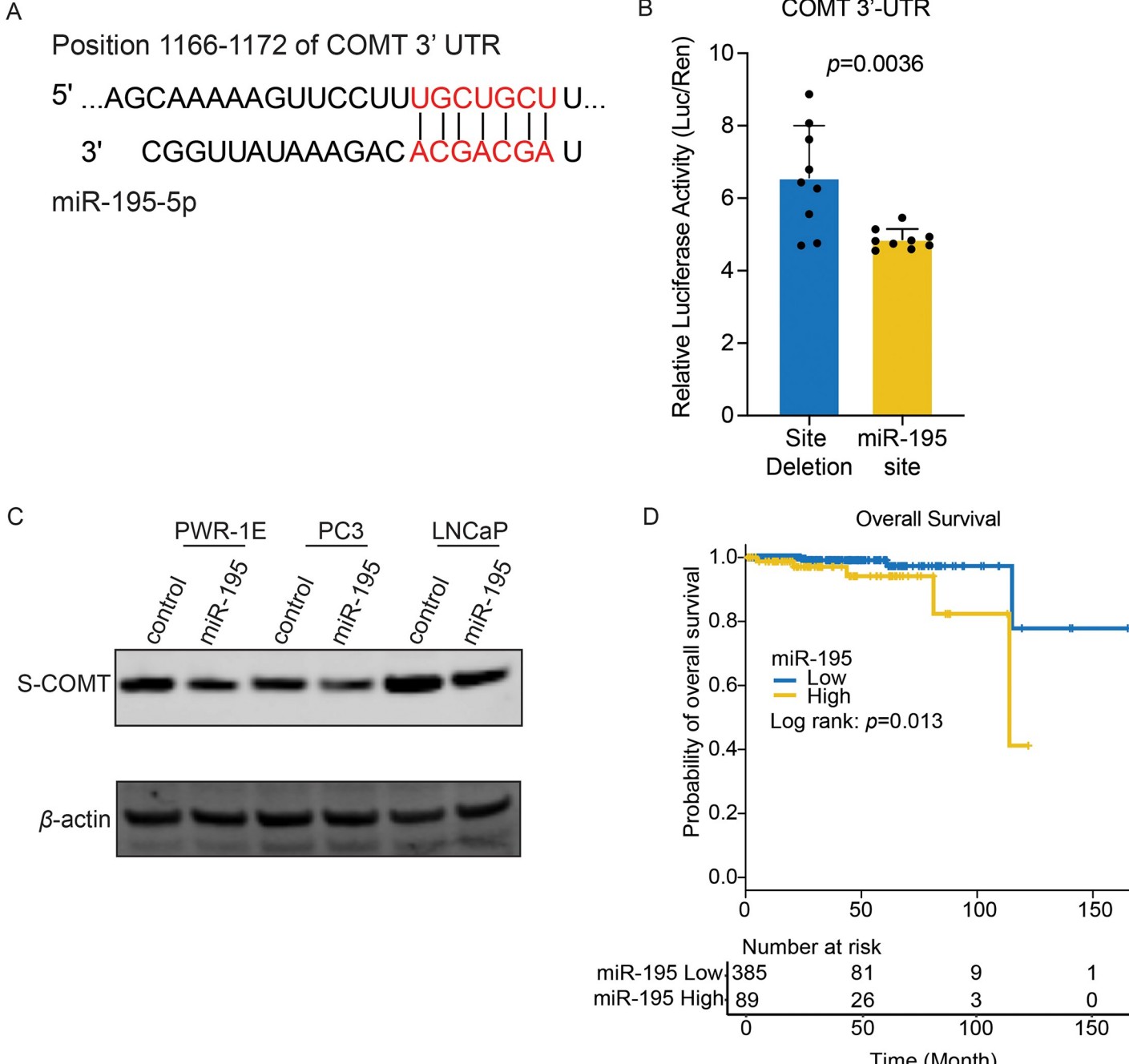

**Fig 6. Regulation of COMT by miR-195.** (A) MiR-195 binding site on the COMT gene 3'-UTR were predicted by TargetScan. (B) Luciferase assay show miRNA-195 capable of targeting COMT. Luciferase activity of constructs of COMT sequences consisting of miR-195 binding site or its deletion were compared in the presence of miR-195 mimics. (C) Overexpressing miR-195 with mimic decreased COMT expression in PWR-1E, LNCaP and PC3 cells by western blot. $\beta$-actin used as loading control. (D) Overall survival analysis of miR-195 shows significantly poor survival rate in miR-195 high expression group (Log rank test: $p$ = 0.013). Data are presented as mean ± SD. The $p$-value is based on paired t-test in B.

observed COMT in the cytoplasm and nucleus of normal and inflamed prostate glandular cells. Nuclear COMT was frequently observed in inflamed prostate epithelial cells in BPH samples, and COMT was also localized in the basal cells. Interestingly in mouse model, basal cells were reported capable of transforming into luminal cells by inflammation and lead the prostate

carcinogenesis process [30]. Another report also demonstrated the ability of basal cells to transform into human prostate adenocarcinoma. These transformed cells can arise via phenotypically luminal cancer cells [31]. Thus, nuclear COMT might be induced by excess catechol-estrogens in basal cells and protect from basal cell originated carcinogenesis as well as in luminal cells.

Proper levels of COMT would thus be essential to prevent the damaging effects of catechol-estrogens in the cells of the prostate. This property is supported by the lower levels of COMT observed in prostate cancer compared to normal regions according to TCGA though a trend ($p = 0.2$; Fig 1A) but significantly compared to BPH in our clinical specimens ($p = 0.004$; Fig 1E). In fact, categorization based on progression in TCGA show advanced stages to be significantly downregulated for COMT ($p = 0.007$) and lower expression resulted in poorer overall survival. In agreement with our results, Singh et al [32] observes COMT to be expressed in benign prostate. This trend of expression is also shown in other cancers. In kidney, both tissues and cell lines have much lower COMT levels in cancer compared to normal counterparts [33]. In breast, Singh et al [34] observed cancer cases to have lower expression of COMT RNA compared to control tissue. For ovary, cancer lines had much lower COMT protein expression compared to ovarian surface epithelial cells [35]. Also, lower COMT expression led to reduced survival time and overall survival for patients with pancreatic cancer [36].

In addition to expression levels in tissue, COMT was also observed downregulated in AR negative cell lines and especially DuPro and DU145 which were minimal to undetectable. Thus, these cell lines are ideal to determine functional effects of COMT by gene re-expression, which has never been previously reported. By expressing COMT protein in cells, we observed reduced migration ability and increased apoptosis of these cancer lines. Also, proliferation was decreased when challenged with mutagenic 4-hydroxyestradiol. A similar effect was observed *in vivo* as COMT-expressing cells inhibited xenograft tumor growth. In concordance, by silencing COMT in LNCaP cells, we observed decreased apoptosis and accelerated migration capability and a recent report by Tolba et al [37] also showed COMT silencing could enhance proliferation of PC3 cells. This tumor suppressive effect of COMT is also shown in other cancers such as pancreatic [36] and colorectal [38]; and in a prior study done in our laboratory, COMT significantly reduced proliferation of renal cancer cells treated with 4-hydroxyestradiol as compared to vector control [33]. These pro-apoptotic and anti-proliferative effects of COMT could therefore contribute to reducing cancer cell growth.

COMT was thus observed to increase apoptosis of cells. This apoptotic activity was associated with a perceived increase in cleaved CASP3 and BID and is further supported by a decrease of anti-apoptotic XIAP and cIAP2. In other cancers, COMT has also been shown to have a pro-apoptotic effect although the genes involved differ. Wu et al [36] found COMT overexpression to cause enhanced apoptosis of gemcitabine-treated pancreatic cancer cells and this was due to enhanced levels of pro-apoptotic Bim and Bax, but reduced anti-apoptotic p-Bad. In 4-hydroxyestradiol stimulated renal cancer cells, COMT caused increased apoptosis along with increased GADD45a levels [33]. Aside from apoptosis-related genes, COMT can play a protective role by affecting other pathway genes such as ER-α, p21cip1, p27kip1, NF-κB (P65) and CYP19A1 in prostate cancer [37], cause a reduction in p-AKT, mutant p53, and cyclin D1 but an increase in p-GSK3-b, PTEN and E-cadherin in pancreatic cancer cells [36], and cause inhibition of p-AKT but increased PTEN, p53, p27, and E-Cadherin in colorectal cancer cells [38].

Lastly, we determined the regulation of COMT in cells. miRNAs are known to inhibit gene expression post-transcriptionally by binding to the 3'UTR of mRNAs [39] and after a search of databases and evaluating several miRNAs, we found miR-195 to be capable of binding to and downregulating COMT. In concordance, miR-195 was observed to be upregulated in prostate

cancer compared to both normal according to TCGA, and to BPH in our clinical specimens wherein COMT was downregulated in cancer. These results thus indicate miR-195 to interact and regulate COMT expression levels. Studies on miRNA regulation of COMT is highly limited in cancer and in the only known report, a polymorphism in the 3'UTR region of COMT could significantly reduce the binding of miR-22 in colon cancer cells [40].

## Conclusions

In summary, our results are the first to show the COMT gene to trigger cell death in prostate cancer by enhancing apoptosis through its pathway genes. Also, that COMT is regulated by miR-195. We conclude that COMT plays a protective role in prostate cancer cells and thus, could be a potential gene of interest for therapeutic development for this cancer as well as a possible biomarker.

## Supporting information

**S1 Fig. Raw images.**
(PDF)

**S2 Fig. COMT expression was detected in human normal prostate glandular cells.**
(PDF)

**S3 Fig. The magnified IHC images in Fig 1D.**
(PDF)

**S4 Fig. COMT knockdown reduced apoptosis and increased migration in LNCaP cell.**
(PDF)

## Acknowledgments

We appreciate Dr. Shahana Majid for her support and assistance with this study.

## Author Contributions

**Conceptualization:** Yutaka Hashimoto, Marisa Shiina, Shigekatsu Maekawa, Taku Kato, Priyanka Kulkarni, Pritha Dasgupta, Yuichiro Tanaka.

**Data curation:** Yutaka Hashimoto, Shigekatsu Maekawa, Varahram Shahryari, Priyanka Kulkarni, Soichiro Yamamura, Sharanjot Saini, Z. Laura Tabatabai, Rajvir Dahiya, Yuichiro Tanaka.

**Formal analysis:** Yutaka Hashimoto, Marisa Shiina, Shigekatsu Maekawa, Varahram Shahryari, Priyanka Kulkarni, Pritha Dasgupta, Z. Laura Tabatabai.

**Funding acquisition:** Yuichiro Tanaka.

**Investigation:** Yutaka Hashimoto, Marisa Shiina, Shigekatsu Maekawa, Varahram Shahryari, Priyanka Kulkarni, Pritha Dasgupta.

**Methodology:** Yutaka Hashimoto.

**Supervision:** Yutaka Hashimoto, Yuichiro Tanaka.

**Validation:** Z. Laura Tabatabai.

**Visualization:** Yutaka Hashimoto, Taku Kato, Soichiro Yamamura, Sharanjot Saini, Rajvir Dahiya.

**Writing – original draft:** Yuichiro Tanaka.

**Writing – review & editing:** Yutaka Hashimoto, Marisa Shiina, Shigekatsu Maekawa, Taku Kato, Varahram Shahryari, Priyanka Kulkarni, Pritha Dasgupta, Soichiro Yamamura, Sharanjot Saini, Z. Laura Tabatabai, Rajvir Dahiya, Yuichiro Tanaka.

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
