## [Decision Letter · Decision Letter 0]

3 Feb 2021

PONE-D-20-40576

Suppressor effect of catechol-O-methyltransferase gene in prostate cancer

PLOS ONE

Dear Dr. Tanaka,

Thank you for submitting your manuscript to PLOS ONE. After careful consideration, we feel that it has merit but does not fully meet PLOS ONE’s publication criteria as it currently stands. Therefore, we invite you to submit a revised version of the manuscript that addresses the points raised during the review process.

We look forward to receiving your revised manuscript.

Kind regards,

Zoran Culig

Academic Editor

PLOS ONE

Additional Editor Comments:

The authors should repeat the key experiments in androgen receptor-positive cells and improve quality of images in the manuscript. They should also clarify whether overexpression of COMT has functional implicatons. 

Journal Requirements:

2.) PLOS ONE now requires that authors provide the original uncropped and unadjusted images underlying all blot or gel results reported in a submission’s figures or Supporting Information files. This policy and the journal’s other requirements for blot/gel reporting and figure preparation are described in detail at https://journals.plos.org/plosone/s/figures#loc-blot-and-gel-reporting-requirements and https://journals.plos.org/plosone/s/figures#loc-preparing-figures-from-image-files. When you submit your revised manuscript, please ensure that your figures adhere fully to these guidelines and provide the original underlying images for all blot or gel data reported in your submission. See the following link for instructions on providing the original image data: https://journals.plos.org/plosone/s/figures#loc-original-images-for-blots-and-gels.

3.) We note that you have stated that you will provide repository information for your data at acceptance. Should your manuscript be accepted for publication, we will hold it until you provide the relevant accession numbers or DOIs necessary to access your data. If you wish to make changes to your Data Availability statement, please describe these changes in your cover letter and we will update your Data Availability statement to reflect the information you provide.

4.) Thank you for submitting the above manuscript to PLOS ONE. During our internal evaluation of the manuscript, we found significant text overlap between your submission and the following previously published works, some of which you are an author.

- https://doi.org/10.18632/oncotarget.2315

- https://doi.org/10.18632/oncotarget.9470

Please revise the manuscript to rephrase the duplicated text, cite your sources, and provide details as to how the current manuscript advances on previous work. Please note that further consideration is dependent on the submission of a manuscript that addresses these concerns about the overlap in text with published work.

Reviewers' comments:

Reviewer's Responses to Questions

**Comments to the Author**

1. Is the manuscript technically sound, and do the data support the conclusions?

Reviewer #1: Partly

Reviewer #2: No

2. Has the statistical analysis been performed appropriately and rigorously? 

Reviewer #1: I Don't Know

Reviewer #2: Yes

3. Have the authors made all data underlying the findings in their manuscript fully available?

Reviewer #1: Yes

Reviewer #2: Yes

4. Is the manuscript presented in an intelligible fashion and written in standard English?

Reviewer #1: Yes

Reviewer #2: Yes

5. Review Comments to the Author

Reviewer #1: The manuscript by Hashimoto and colleagues describes suppressive effect of catechol-O-methyltransferase (COMT) gene in prostate cancer. The TCGA and immunohistochemical analysis showed COMT down-regulation in prostate cancer and cancer cell lines. Transfection of COMT suppressed prostate cancer cell migration and enhanced apoptosis as well as suppressed proliferation in culture in the presence of 4-hydroxyestradiol and in xenograft tumors. Mechanistic analysis discovered regulation of COMT by miR-195. These findings are novel and have potential clinical relevance. However, the authors will need to address following concerns.

1. In Fig. 2A, is the COMT overexpression level physiologically relevant?

2. Does COMT knockdown affect proliferation, migration, and/or apoptosis of prostatic cells?

3. In this study, AR-negative cells were used. However, prostate cancer cells in the clinical specimens are in general AR-positive. The authors should also use AR-positive prostate cancer cells.

4. In Fig. 6B, the authors showed miR-195 suppression of COMT expression via its 3’-UTR using luciferase assay. It will be more convincing to show that miR-195 knockdown and/or overexpression can indeed modulate the endogenous COMT expression.

Reviewer #2: The manuscript by Hashimoto et al explores expression of COMT in prostate cancer and BPH and a further mechanism for silencing of COMT in prostate cancer due to miR-195.

Introduction:

First sentence “…for men during senescence…” - rephrase as “with aging” or “among aging men” or something similar, for clarity.

Materials and Methods:

Include a reference for DuPro cells in the materials and methods as this is not a commonly used line.

Please include IHC assay positive and negative controls (cells or tissues validated via a different means to contain and not contain COMT).

Results:

Figure 1 – it would also be of interest to see COMT analyzed in relation to Gleason grade.

It’s unclear why BPH was used as a comparison to prostate cancer. Benign, normal appearing prostate in cases with prostate cancer would have been more informative, and should be included if possible. The IHC images provided in Figure 1D are not at a high enough power (along with low resolution images) to see what the staining pattern is. It looks like possibly some (or all?) of the staining is basal cell staining which obviously would not be applicable to prostate cancer, but it’s impossible to tell with the images presented. Higher power images should be included. Ideally, IHC images of normal prostate (e.g. normal appearing benign regions from radical prostatectomy) should be included.

If forced expression on COMT triggers apoptosis, this alone could explain the reduced migration and lack of growth in xenograft studies. It’s not really “suppression of tumor growth” so much as triggering of cell death? Please revise the conclusions based on these assays accordingly.

Line 344/345 – change “significantly oncogenic” to upregulated in cancer compared to normal prostate (or something similar, for clarity).

Conclusions (also in abstract) – why would COMT be a “therapeutic target” in prostate cancer if it is absent in prostate cancer?

6. PLOS authors have the option to publish the peer review history of their article (what does this mean?). If published, this will include your full peer review and any attached files.

Reviewer #1: No

Reviewer #2: No

---

## [Author Response · Author response to Decision Letter 0]

23 Apr 2021

Point by point response to reviewer’s comments

Additional Editor Comments: The authors should repeat the key experiments in androgen receptor-positive cells and improve quality of images in the manuscript. They should also clarify whether overexpression of COMT has functional implications.

Response: We evaluated COMT levels in AR+ LNCaP cells and observed positive COMT expression (Figure 1F,G). Since COMT was expressed, we therefore performed COMT knockdown instead and after evaluating functional effects, we observed results that were consistent with COMT overexpression (Figure S2). We also added better quality images of immunohistochemical results (Figure 1E) and provide an expanded region of staining in Figure S1. 

Reviewer #1

Comment 1: In Fig. 2A, is the COMT overexpression level physiologically relevant?

Response: We also performed COMT knockdown in LNCaP cells to evaluate physiological relevance, see S2 Figure.

Comment 2: Does COMT knockdown affect proliferation, migration, and/or apoptosis of prostatic cells?

Response: We added new S2 Figure that investigated the impact COMT knockdown has on apoptosis and migration of LNCaP cells.

Comment 3: In this study, AR-negative cells were used. However, prostate cancer cells in the clinical specimens are in general AR-positive. The authors should also use AR-positive prostate cancer cells.

Response: We have measured COMT levels in AR+ PCa cells (LNCaP) and they showed positive COMT expression. For this reason, we could not perform overexpression using AR+ cells. Thus in these LNCaP cells, we now show effects of COMT silencing on functional effects in S2 Figure.

Comment 4: In Fig. 6B, the authors showed miR-195 suppression of COMT expression via its 3’-UTR using luciferase assay. It will be more convincing to show that miR-195 knockdown and/or overexpression can indeed modulate the endogenous COMT expression.

Response: We performed miR-195 overexpression by using miR-195 mimic in PWR1E, LNCaP and PC3 cells and confirmed that miR-195 mimic could decrease COMT expression, see Figure 6C.

Reviewer #2

Comment 1: First sentence “…for men during senescence…” - rephrase as “with aging” or “among aging men” or something similar, for clarity.

Response: We thank this comment. We edited lines 44-49 since the editorial office also requested some modification on these sentences.

Comment 2: Include a reference for DuPro cells in the materials and methods as this is not a commonly used line.

Response: A new reference entitled, “Establishment and characterization of a new human prostatic carcinoma cell line (DuPro-1)”, was added as ref. #20 in this version. See line 103.

Comment 3: Please include IHC assay positive and negative controls (cells or tissues validated via a different means to contain and not contain COMT).

Response: Thank you very much for pointing this out. We did not clearly mention but we used same antibody (Abcam, #ab185954) for western blotting and IHC. Thus, each negative and positive control would be DU145 and PWR1E in Fig 1G. The detailed product information was added in lines 120 and 150 to reconcile the confusion. Also, I would like to point out that picture #1 in Fig 1E was replaced with the sample that shows the highest COMT expression level in BPH samples. This picture #1 is an example of positive COMT staining with a score value of 38.6 whereas picture #4 is an example of negative staining with a score value of 0. We performed IHC because detecting protein is informative whereas measuring longer mRNAs are challenging using FFPE samples.

Comment 4: Figure 1 – it would also be of interest to see COMT analyzed in relation to Gleason grade.

Response: We added a new boxplot in Fig 1C. We divided into ≤GS3+4 versus ≥GS4+3 which gave a more even n size using TCGA PRAD samples. We found that COMT expression levels are slightly reduced in higher GS compared to lower GS group.

Comment 5: It’s unclear why BPH was used as a comparison to prostate cancer. Benign, normal appearing prostate in cases with prostate cancer would have been more informative, and should be included if possible. The IHC images provided in Figure 1D are not at a high enough power (along with low resolution images) to see what the staining pattern is. It looks like possibly some (or all?) of the staining is basal cell staining which obviously would not be applicable to prostate cancer, but it’s impossible to tell with the images presented. Higher power images should be included. Ideally, IHC images of normal prostate (e.g. normal appearing benign regions from radical prostatectomy) should be included.

Response: As per reviewer’s comment, we replaced images with higher resolution images and observed COMT staining in luminal cells and basal region. and rephrased the sentence (line 239). Also, magnified images are included in Figure S1. Since the merged PDF file is using lower resolution, please click the link “Click here to access/download;Figure;FigX.tif” at the corner of figure page and the original higher resolution files will appear. Unfortunately, normal region FFPE samples were not available and these are the pictures that we have.

Comment 6: If forced expression on COMT triggers apoptosis, this alone could explain the reduced migration and lack of growth in xenograft studies. It’s not really “suppression of tumor growth” so much as triggering of cell death? Please revise the conclusions based on these assays accordingly.

Response: Thank you very much for this comment. We modified sentences, see line 430.

Comment 7: Line 344/345 – change “significantly oncogenic” to upregulated in cancer compared to normal prostate (or something similar, for clarity).

Response: We replace this phrase along with reviewer’s comment. See line 333.

Comment 7: Conclusions (also in abstract) – why would COMT be a “therapeutic target” in prostate cancer if it is absent in prostate cancer?

Response: Appreciated for pointing this error out. We removed “target” and replace with possible gene for therapeutic development. See line 41 and 432.

---

## [Decision Letter · Decision Letter 1]

16 Jun 2021

PONE-D-20-40576R1

Suppressor effect of catechol-O-methyltransferase gene in prostate cancer

PLOS ONEPlease ensure that your decision is justified on PLOS ONE’s publication criteria and not, for example, on novelty or perceived impact.

Dear Dr. Tanaka,

Thank you for submitting your manuscript to PLOS ONE. After careful consideration, we feel that it has merit but does not fully meet PLOS ONE’s publication criteria as it currently stands. Therefore, we invite you to submit a revised version of the manuscript that addresses the points raised during the review process.

=======The authors should pay attention to images as stated in one of our reviews. Otherwise, the paper contains several improvements. 

We look forward to receiving your revised manuscript.

Kind regards,

Zoran Culig

Academic Editor

PLOS ONE

Journal Requirements:

Additional Editor Comments (if provided):

Reviewers' comments:

Reviewer's Responses to Questions

**Comments to the Author**

1. If the authors have adequately addressed your comments raised in a previous round of review and you feel that this manuscript is now acceptable for publication, you may indicate that here to bypass the “Comments to the Author” section, enter your conflict of interest statement in the “Confidential to Editor” section, and submit your "Accept" recommendation.

Reviewer #1: All comments have been addressed

Reviewer #2: (No Response)

2. Is the manuscript technically sound, and do the data support the conclusions?

Reviewer #1: Yes

Reviewer #2: No

3. Has the statistical analysis been performed appropriately and rigorously? 

Reviewer #1: Yes

Reviewer #2: Yes

4. Have the authors made all data underlying the findings in their manuscript fully available?

Reviewer #1: Yes

Reviewer #2: Yes

5. Is the manuscript presented in an intelligible fashion and written in standard English?

Reviewer #1: Yes

Reviewer #2: Yes

6. Review Comments to the Author

Reviewer #1: The authors have addressed previous critiques adequately and provided additional data. The authors have improved the manuscript in the revision.

Reviewer #2: The response to this reviewer's comments are very much appreciated however there is one critical point that has not been adequately addressed. The IHC images shown in Fig. 1E and S1 (which is labeled as S2 in the figure) are much clearer (thank you) but appear to demonstrate that COMT may be limited to prostate atrophy and basal cells and is completely absent in normal luminal prostate epithelium and prostate cancer. The cancer region magnified in Fig S1 does not demonstrate any cancer cell staining. The authors need to provide IHC evidence that COMT is actually expressed in normal appearing luminal epithelium and (presumably to a lesser extent) in prostate cancer cells if that is indeed the case as much of the premise of this manuscript is based on this.

7. PLOS authors have the option to publish the peer review history of their article (what does this mean?). If published, this will include your full peer review and any attached files.

Reviewer #1: No

Reviewer #2: No

---

## [Author Response · Author response to Decision Letter 1]

9 Jul 2021

Reviewer #2: 

COMMENT: The response to this reviewer's comments are very much appreciated however there is one critical point that has not been adequately addressed. The IHC images shown in Fig. 1E and S1 (which is labeled as S2 in the figure) are much clearer (thank you) but appear to demonstrate that COMT may be limited to prostate atrophy and basal cells and is completely absent in normal luminal prostate epithelium and prostate cancer. The cancer region magnified in Fig S1 does not demonstrate any cancer cell staining. The authors need to provide IHC evidence that COMT is actually expressed in normal appearing luminal epithelium and (presumably to a lesser extent) in prostate cancer cells if that is indeed the case as much of the premise of this manuscript is based on this.

RESPONSE: Thank you very much for your comment. As mentioned before, we have only samples donated from patients who went through radical prostatectomy. Thus, we could not obtain the normal tissue samples. Instead, we provided the link and a picture from the database Human Protein Atlas that displays human normal prostate tissues stained with COMT antibodies. COMT expression was detected in the cytoplasm and nucleus of glandular cells (luminal and basal cells). We now add arrows in Figure S1 which point to COMT-positivity in luminal cells, as seen in tissues in Protein Atlas. In the Discussion section, we discuss this matter and also add references that demonstrate basal cell transformation to be involved in prostate carcinogenesis (lines 385-393).

---

## [Editor Report · Decision Letter 2]

2 Aug 2021

Suppressor effect of catechol-O-methyltransferase gene in prostate cancer

PONE-D-20-40576R2

Dear Dr. Tanaka,

We’re pleased to inform you that your manuscript has been judged scientifically suitable for publication and will be formally accepted for publication once it meets all outstanding technical requirements.

Kind regards,

Zoran Culig

Academic Editor

PLOS ONE
---

## [Editor Report · Acceptance letter]

20 Sep 2021

PONE-D-20-40576R2 

Suppressor effect of catechol-O-methyltransferase gene in prostate cancer 

Dear Dr. Tanaka:

I'm pleased to inform you that your manuscript has been deemed suitable for publication in PLOS ONE. Congratulations! Your manuscript is now with our production department. 

Kind regards, 

on behalf of

Dr. Zoran Culig 

Academic Editor

PLOS ONE